# BlockSwap: Fisher-guided Block Substitution for Network Compression on a Budget

**Jack Turner,**\* **Elliot J. Crowley,**\* **Michael O'Boyle, Amos Storkey, Gavin Gray**†
School of Informatics
University of Edinburgh
`{jack.turner,elliot.j.crowley}@ed.ac.uk, mob@inf.ed.ac.uk, {a.storkey,g.d.b.gray}@ed.ac.uk`

## Abstract

The desire to map neural networks to varying-capacity devices has led to the development of a wealth of compression techniques, many of which involve replacing standard convolutional blocks in a large network with cheap alternative blocks. However, not all blocks are created equally; for a required compute budget there may exist a potent combination of many different cheap blocks, though exhaustively searching for such a combination is prohibitively expensive. In this work, we develop *BlockSwap*: a fast algorithm for choosing networks with interleaved block types by passing a single minibatch of training data through randomly initialised networks and gauging their *Fisher potential*. These networks can then be used as students and distilled with the original large network as a teacher. We demonstrate the effectiveness of the chosen networks across CIFAR-10 and ImageNet for classification, and COCO for detection, and provide a comprehensive ablation study of our approach. BlockSwap quickly explores possible block configurations using a simple architecture ranking system, yielding highly competitive networks in orders of magnitude less time than most architecture search techniques (e.g. under 5 minutes on a single GPU for CIFAR-10). Code is available at `https://github.com/BayesWatch/pytorch-blockswap`.

## 1 Introduction

Deep Convolutional Neural Networks are extremely popular, and demonstrate strong performance on a variety of challenging tasks. Because of this, there exist a large range of scenarios in the wild in which practitioners wish to deploy these networks e.g. pedestrian detection in a vehicle's computer, human activity recognition with wearables (Radu et al., 2018).

These networks are largely over-parameterised; a fact that can be exploited when specialising networks for different resource budgets. For example, there is a wealth of work demonstrating that it is possible to replace expensive convolutional blocks in a large network with cheap alternatives e.g. those using grouped convolutions (Chollet, 2017; Xie et al., 2017; Ioannou et al., 2017; Huang et al., 2018) or bottleneck structures (He et al., 2016; Sandler et al., 2018; Peng et al., 2018). This creates a smaller network which may be used as a *student*, and trained through distillation (Ba & Caruana, 2014; Hinton et al., 2015) with the original large network as a teacher to retain performance.

Typically, each network block is replaced with a single cheap alternative, producing a single-blocktype network for a given budget. By cheapening each block equally, one relies on the assumption that each block is of equal importance. We posit instead that for each budget, there exist more powerful *mixed-blocktype networks* that assign non-uniform importance to each block by cheapening them to different extents. We now have a paradox of choice; for a given budget, it is not obvious *which* cheap alternatives to use for our student network, nor *where* to place them.

Let's assume we have a large network that consists of $B$ expensive convolutional blocks, and a candidate pool of $C$ cheap blocks that we could substitute each of these out for, and we are given a limit on the number of parameters (and therefore memory) that the network can use. Which of

---

\*Equal contribution.
†Now at the University of Toronto.

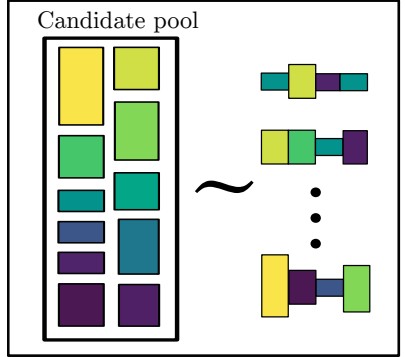 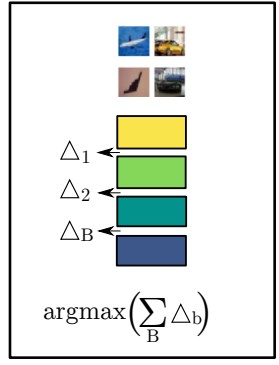 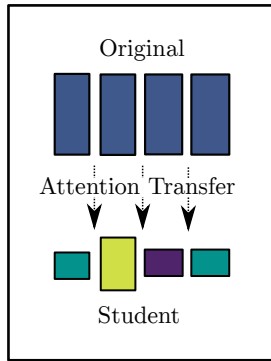

| 1. Sample random architectures | 2. Rank by Fisher potential after one minibatch | 3. Train with AT |

Figure 1: The three-step BlockSwap pipeline. Beginning with a large network, we sample a list of candidate architectures by replacing its blocks with cheap alternatives. In Step 2, we rank each candidate by its Fisher potential after a single minibatch of training data. In Step 3, we select the highest ranked architecture and train it with attention transfer from the original network.

the $C^B$ *mixed-blocktype* networks is appropriate? It rapidly becomes infeasible to exhaustively consider these networks, even for single digit $C$. We could turn to neural architecture search (NAS) techniques (Zoph & Le, 2017; Zoph et al., 2018; Pham et al., 2018; Tan et al., 2019; Wu et al., 2019; Liu et al., 2019a) but these introduce a gigantic computational overhead; even one of the fastest such approaches (Pham et al., 2018) still requires half a day of search time on a GPU. Moreover, in real-world scenarios, allocated budgets can change quickly; we would like to be able to find a good network in a matter of minutes, not hours or days.

Our goal in this paper is, given a desired parameter budget, to quickly identify a suitable *mixed-blocktype* version of the original network that makes for a powerful student. We present a simple method—*BlockSwap*—to achieve this. First, we randomly sample a collection of candidate *mixed-blocktype* architectures that satisfy the parameter budget. A single minibatch is then pushed through each candidate network to calculate its *Fisher potential*: the sum of the total (empirical) Fisher information for each of its blocks. Finally, the network with the highest potential is selected as a student and trained through the distillation method of attention transfer (Zagoruyko & Komodakis, 2017) with the original teacher. Our method is illustrated in Figure 1.

In Section 3 we describe the block substitutions used in BlockSwap, distillation via attention transfer, and Fisher information. We elaborate on our method in Section 4 as well as providing a comprehensive ablation study. Finally, we experimentally verify the potency of BlockSwap on CIFAR-10 (Section 5) as well as ImageNet (Section 6) and COCO (Section 7). Our contributions are as follows:

1. We introduce BlockSwap, an algorithm for reducing large neural networks by performing block-wise substitution. We show that this outperforms other top-down approaches such as depth/width scaling, parameter pruning, and random substitution.

2. We outline a simple method for quickly evaluating candidate models via Fisher information, which matches the performance of bottom-up approaches while reducing search time from days to minutes.

3. We conduct ablation studies to validate our methodology, highlighting the benefits of block mixing, and confirming that our ranking metric is highly correlated to the final error.

## 2   RELATED WORK

Neural networks tend to be overparameterised: Denil et al. (2013) accurately predict most of the weights in a network from a small subset; Frankle & Carbin (2019) hypothesise that within a large network, there exists a fortuitously initialised subnetwork that drives its performance. However, it remains difficult to exploit this overparameterisation without taking a hit in performance.

One means to combat this is to use a large *teacher* network to regularise the training of a small *student* network; a process known as distillation. The small network is trained from scratch, but is also forced to match the outputs (Ba & Caruana, 2014; Hinton et al., 2015) or activation statistics (Romero et al., 2015; Zagoruyko & Komodakis, 2017; Kim et al., 2018) of the teacher using an additional loss term. When utilising distillation one must decide how to create a student network. A simple approach would be to reduce the depth of the original large network, although this can prove detrimental (Urban et al., 2017). An effective strategy is to create a student by replacing all the teacher's convolutions with grouped alternatives (Crowley et al., 2018a).

Grouped convolutions are a popular replacement for standard convolutions as they drastically cut the number of parameters used by splitting the input along the channel dimension and applying a much cheaper convolution on each split. They were originally used in AlexNet (Krizhevsky et al., 2012) due to GPU memory limitations, and have appeared in several subsequent architectures (Ioffe & Szegedy, 2015; Chollet, 2017; Xie et al., 2017; Ioannou et al., 2017; Huang et al., 2018). However, as the number of groups increases, fewer channels are mixed, which hinders representational capacity. MobileNet (Howard et al., 2017) compensates for this by following its heavily-grouped depthwise convolutions by a pointwise ($1 \times 1$) convolution to allow for channel mixing.

The increasing complexity of neural network designs has encouraged the development of methods for automating *neural architecture search* (NAS). Zoph et al. (2018) use an RNN to generate network block descriptions and filter the options using reinforcement learning. These blocks are stacked to form a full neural network. This is an extremely expensive process, utilising **450 GPUs** over the course of **3 days**. To address this, Pham et al. (2018) propose giving all models access to a shared set of weights, achieving similar performance to Zoph et al. (2018) with a single GPU in less than 24 hours. Subsequent works have made extensive use of this technique (Liu et al., 2019a; Luo et al., 2018; Chen et al., 2018b). However, it has been observed that under the constrained architecture search space of the above methods, random architecture search provides a competitive baseline (Li & Talwalkar, 2019; Yu et al., 2020). In particular, Yu et al. (2020) show that weight sharing hampers the ability of candidate networks to learn and causes many NAS techniques to find suboptimal architectures.

NAS techniques predominantly take a bottom-up approach; they find a powerful building block, and form neural networks using stacks of these blocks. Other works have taken a top-down approach to find architectures using pruning (Chen et al., 2018a; Lee et al., 2019; Liu et al., 2019b; Crowley et al., 2018b; Frankle & Carbin, 2019). Chen et al. (2018a) take a pre-trained network and apply a principled framework to compress it under operational constraints such as latency. In SNIP (Lee et al., 2019), one randomly initialises a large network and quantifies the sensitivity of each connection using gradient magnitudes. The lowest sensitivity connections are removed to produce a sparse architecture which is then trained as normal.

## 3 PRELIMINARIES

### 3.1 SUBSTITUTE BLOCKS

Here, we will briefly elaborate on the block substitutions used for *BlockSwap*. A tabular comparison is given in Appendix A. The block choices are deliberately simplistic. We can therefore demonstrate that it is the *combination* of blocks that is important rather than the representational capacity of a specific highly-engineered block e.g. one from the NAS literature.

The blocks considered are variations of the standard block used in residual networks. In the majority of these blocks, the input has the same number of channels as the output, so we describe their parameter cost assuming this is the case. This standard block contains two convolutional layers, each using $N$ lots of $N \times k \times k$ filters where $k$ is the kernel size. Assuming the costs of batch-norm (BN) layers and shortcut convolutions (where applicable) are negligible, the block uses a total of $2N^2k^2$ parameters.

**Grouped+Pointwise Block – G(g)**   In a grouped convolution, the $N$ channel input is split along the channel dimension into $g$ groups, each of which has $N/g$ channels. Each group goes through its own convolution which outputs $N/g$ channels, and all the outputs are concatenated along the channel dimension. This uses $g \times (N/g)^2k^2 = (N^2k^2)/g$ parameters. To compensate for reduced

channel mixing, each convolution is further followed by a pointwise $(1 \times 1)$ convolution, incurring an extra cost of $N^2$. For this block, each full convolution has been replaced with a grouped+pointwise convolution, and so the block uses $2((N^2k^2)/g + N^2)$ parameters.

**Bottleneck Block – B(b)**  In this block, a pointwise convolution is used to reduce the number of channels of the input by a factor of $b$ before a standard convolution is applied. Then, another pointwise convolution brings the channel size back up. This uses $(N/b)^2k^2 + 2N^2/b$ parameters.

**Bottleneck Grouped+Pointwise Block – BG(b, g)**  This is identical to the bottleneck block, except the standard convolution is further split into $g$ groups, and so uses $(N/bg)^2k^2 + 2N^2/b$ parameters.

## 3.2 DISTILLATION VIA ATTENTION TRANSFER

Attention transfer (Zagoruyko & Komodakis, 2017) is a distillation technique whereby a student network is trained such that its *attention maps* at several distinct *attention points* are made to be similar to those produced by a large teacher network. A formal definition follows: Consider a choice of layers $i = 1, 2, ..., L$ in a teacher network with $L$ layers, and the corresponding layers in the student network. At each chosen layer $i$ of the teacher network, collect the spatial map of the activations for channel $j$ into the vector $\mathbf{a}_{ij}^t$. Let $A_i^t$ collect $\mathbf{a}_{ij}^t$ for all $j$. Likewise for the student network we correspondingly collect into $\mathbf{a}_{ij}^s$ and $A_i^s$. Now given some choice of mapping $\mathbf{f}(A_i)$ that maps each collection of the form $A_i$ into a vector, attention transfer involves learning the student network by minimising

$$\mathcal{L}_{AT} = \mathcal{L}_{CE} + \beta \sum_{i=1}^{L} \left\| \frac{\mathbf{f}(A_i^t)}{||\mathbf{f}(A_i^t)||_2} - \frac{\mathbf{f}(A_i^s)}{||\mathbf{f}(A_i^s)||_2} \right\|_2, \tag{1}$$

where $\beta$ is a hyperparameter, and $\mathcal{L}_{CE}$ is the standard cross-entropy loss. In Zagoruyko & Komodakis (2017) the authors use $\mathbf{f}(A_i) = (1/N_{A_i}) \sum_{j=1}^{N_{A_i}} \mathbf{a}_{ij}^2$, where $N_{A_i}$ is the number of channels at layer $i$.

## 3.3 FISHER INFORMATION

Theis et al. (2018) derive a second order approximation of the change in loss that would occur on the removal of a particular channel activation in a neural network; they demonstrate that this is equivalent to calculating an empirical estimate of the Fisher information for a binary mask parameter that is used to toggle that channel on or off. They use this signal $\Delta_c$ to identify the least important activation channels, and remove their corresponding weights while pruning. Formally, let us consider a single channel of an activation in a network due to some input minibatch of $N$ examples. Let us denote the values for this channel as $a$: a $N \times W \times H$ tensor where $W$ and $H$ are the channel's spatial width and height. Let us refer to the entry corresponding to example $n$ in the mini-batch at location $(i, j)$ as $a_{nij}$. We can backpropagate the network's loss $\mathcal{L}$ to obtain the gradient of $\mathcal{L}$ with respect to this activation channel $\frac{\partial \mathcal{L}}{\partial a}$. Let us denote this gradient as $g$ and index it as $g_{nij}$. $\Delta_c$ can then be computed by

$$\Delta_c = \frac{1}{2N} \sum_{n}^{N} \left( \sum_{i}^{W} \sum_{j}^{H} a_{nij} g_{nij} \right)^2. \tag{2}$$

In this work, we are interested in obtaining the Fisher information for a whole block. We approximate this quantity by summing $\Delta_c$ for every output channel in a block as

$$\Delta_b = \sum_{c}^{C} \Delta_c. \tag{3}$$

Using an approximation to the Taylor expansion of a change in loss to gauge the saliency of individual parameters originated in LeCun et al. (1989) and has inspired many works in pruning (Hassibi & Stork, 1993; Molchanov et al., 2017; Guo et al., 2016; Srinivas & Babu, 2015) and quantisation (Choi et al., 2017; Hou et al., 2017).

## 4 METHOD

Let us denote a large teacher network $T$ composed of $B$ blocks each of type $S$ as $T = [S_1, S_2, ..., S_B]$. Each of these may be replaced by a cheap block $C_r$, chosen from a list of candidates $C_1, C_2, ..., C_N$ of various representational capacities. We wish to construct a smaller model $M = [C_{r1}, C_{r2}, ..., C_{rB}]$ that is powerful, and within a given parameter budget. But, the space of possible block configurations is very large ($\sim 10^{23}$ in Section 5). Even when using a cheap network evaluation strategy it is not possible to exhaustively search through the available network architectures. What we require is a method to quickly propose and score possible configurations with as little training as possible. We develop BlockSwap to achieve this.

First, we obtain candidate architectures through rejection sampling: we generate mixed-blocktype architecture proposals at random and only save those that satisfy our parameter budget. As these are only proposals, this step does not require instantiating any networks; parameter count can be inferred directly from the proposed configuration, making this a very cheap operation.

Second, we score each of our saved candidates by its *Fisher potential* to determine which network to train. We obtain this score for each candidate as follows: we initialise the network *from scratch* and then place *probes* after the last convolution in each block. A single minibatch of training data is then passed through the network, and the resulting cross-entropy loss is backpropagated. The probe measures the total Fisher information of the block $\Delta_b$ by summing $\Delta_c$ (Equation 2) for each channel in the layer it is placed after. For this step, minibatch size is set equal to the size used during training. We then sum this quantity across all blocks to give us the Fisher potential.

The intuition for this metric is as follows: the Fisher potential is the trace of the Fisher Information matrix of the activations. It is an aggregate of the total information each block contains about the class label (under a simplifying conditional independence assumption). During training it is this information about the class that drives learning, and the initial learning steps are key. Hence higher information values tend to result in higher efficiency block utilisation.

Once we have scored each candidate architecture using the Fisher potential, we select the one with the highest score and train it using attention transfer. The training hyperparameters can be mirrored from $T$. We use the following blocks (defined in Section 3.1) as candidate blocks:

- $B(b)$ for $b \in \{2, 4\}$

- $G(g)$ for $g \in \{2, 4, 8, 16, N/16, N/8, N/4, N/2, N\}$

- $BG(2, g)$ for $g \in \{2, 4, 8, 16, M/16, M/8, M/4, M/2, M\}$

where $N$ is the number of channels in a block, and $M$ is the number of channels after a bottleneck— this is $N/2$ when $b = 2$. We also use the standard block $S$ as a candidate choice, so as to not force blocks to reduce capacity where it is imperative.

### 4.1 WHY MIX BLOCK TYPES?

A reasonable question to ask is whether mixed-blocktype architectures will perform better than the use of a single blocktype. As a simple check for this, we chose parameter budgets of 200K, 400K, and 800K and at each budget, compared single-blocktype models to one hundred randomly assembled, mixed-blocktype models. The models are WideResNets (Zagoruyko & Komodakis, 2016) with substituted blocks trained on CIFAR-10.

In Figure 2 we plot the final test error of the random architectures (blue densities) against their single-blocktype counterparts (the red dotted lines). We observe that for each of the given parameter budgets there exist combinations of mixed blocks that are more powerful than their single blocktype equivalents. However, single-blocktype networks typically sit below the mean of the density of the random networks. This suggests that, though random search has been shown to work well in other architecture search settings (Yu et al., 2020), it will yield sub-optimal structures in this case.

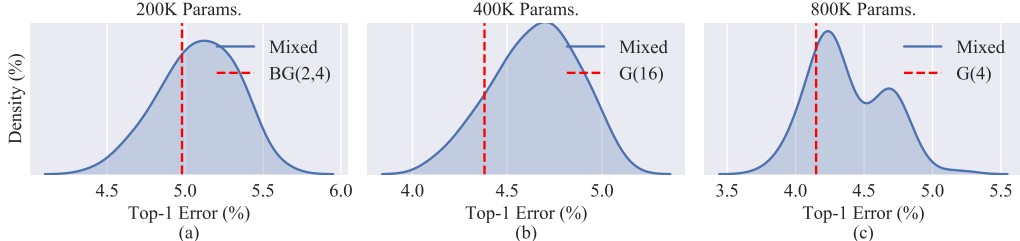

Figure 2: For parameter budgets of (a) 200K, (b) 400K and (c) 800K we train a single blocktype network and 100 random mixed blocktype networks. This lets us examine the distribution of mixed blocktype architectures to verify the existence of performant ones. The dotted red line on each plot represents models composed of a single block type, and the shaded blue density represents randomly constructed, mixed-blocktype networks. For example, (a) shows that a WRN-40-2 with every block substituted for BG(2,4) trains to 5.0% error, while the mean error of randomly constructed networks at the same parameter budget lies at around 5.15% error. This implies that random architecture search will not work here, and that a better strategy is needed.

## 4.2 WHY USE A SINGLE MINIBATCH? HOW DOES YOUR RANKING METRIC COMPARE TO ALTERNATIVES?

Though there are no common metrics for architecture ranking, neural architecture search methods often use information from early learning stages to evaluate possible candidates. In order to assess the fastest, most accurate indicator of final error, we examine our 100 random mixed-blocktype networks for parameter budgets of 200K, 400K, and 800K and record how final error correlates with (i) training accuracy, (ii) total $\ell 2$ norms of weights, (iii) summed absolute gradient norms of weights (Golub et al., 2019), and (iv) Fisher potential after 1, 10, and 100 minibatches of training in Table 1. We found that Fisher potential was by far the most robust metric, and that there were diminishing returns as the number of minibatches increases.

Table 1: Spearman Rank correlation scores for several ranking metrics when ranking 100 random architectures at three parameter budgets. Negative correlation implies that as the metric score goes up, final error goes down. Summing gradient norms follows a similar pattern to Fisher potential but appears to be less robust and moderately less accurate. $\ell 2$-norms are extremely volatile and largely uninformative. Crucially, adding more than 1 minibatch has diminishing returns.

| | 200K Params. | | | 400K Params. | | | 800K Params | | |
|---|---|---|---|---|---|---|---|---|---|
| Minibatches | 1 | 10 | 100 | 1 | 10 | 100 | 1 | 10 | 100 |
| Accuracy | -0.004 | -0.022 | **-0.495** | -0.068 | -0.282 | -0.386 | 0.042 | -0.211 | -0.257 |
| $\ell 2$ norms | -0.083 | -0.185 | -0.289 | 0.093 | -0.213 | -0.251 | 0.368 | 0.226 | 0.140 |
| Grad Norms | -0.541 | -0.490 | 0.326 | -0.612 | -0.444 | 0.402 | -0.608 | 0.340 | **0.558** |
| Fisher | **-0.602** | **-0.621** | -0.439 | **-0.685** | **-0.667** | **-0.508** | **-0.635** | **-0.638** | -0.277 |

## 4.3 HOW MANY SAMPLES ARE NEEDED?

Figure 2 suggests that sampling 100 random architectures will reliably yield at least one mixed-blocktype network that outperforms a similarly parameterised single blocktype network. However, accepting that there is some noise in our ranking metric, we assume that we will need to take more than 100 samples in order to reliably detect powerful architectures. As an illustration, at the budget of 400K parameters a single blocktype alternative has a test error of $4.45\%$, whereas BlockSwap finds networks with final test errors of $4.85\%$. $4.54\%$, and $4.21\%$ after 10, 100, and 1000 samples respectively. We empirically found that 1000 samples was enough to be robust to various parameter budgets on the tasks we considered.

### 4.4 What do "good networks" look like?

For each of the three parameter budgets (200K, 400K, 800K) we inspected the most common block choices for "good" and "bad" networks, where good and bad are networks with final error greater two standard deviations below or above the mean respectively. We found that overwhelmingly, the types of blocks used in both good and bad networks was very similar, implying that **it is the *placement* of the blocks instead of the *types* of the blocks that matters**. We examine this further in Appendix B.

## 5 CIFAR Experiments

Here, we evaluate student networks obtained using *BlockSwap* on the CIFAR-10 image classification dataset (Krizhevsky, 2009). We benchmark these against competing student networks for a range of parameter budgets. To recapitulate, the BlockSwap networks are found by taking 1000 random samples from the space of possible block combinations that satisfy our constraint (in this case, parameter budget). These points are ranked by Fisher potential after a single minibatch of training data, and the network with the highest potential is chosen.

The structure we use is that of a WideResNet (Zagoruyko & Komodakis, 2017), since we can use them to construct compact, high performance networks that are broadly representative of the most commonly used neural architectures. A WideResNet with depth 40, and width multiplier 2—WRN-40-2—is trained and used as a teacher. It consists of 18 blocks and has 2.2 million parameters. A student network is generated and is trained from scratch using attention transfer with the teacher. Our BlockSwap students are WRN-40-2 nets where each of its 18 blocks are determined using our method with the blocks outlined in Section 4. We compare against the following students:

1. Reduced width/depth versions of the teacher: WRN-16-1,WRN-16-2, WRN-40-1

2. Single block-type teacher reductions: every block in the teacher is swapped out for (i) the MBConv6 block from MobileNet v2 (Sandler et al., 2018) and (ii) the (normal cell) block discovered in DARTS (Liu et al., 2019a).

3. Pruned teacher reductions: we compare against (i) the magnitude-based pruning methodology from Han et al. (2016) and (ii) SNIP (Lee et al., 2019) versions of the teacher.

We additionally compare against CondenseNet-86 (Huang et al., 2018), distilled with the Born-Again strategy described by Furlanello et al. (2018).

First, we train three teacher networks independently. These are used to train all of our students; each student network is trained three times, once with each of these teachers. Figure 3 shows the mean test errors of BlockSwap students at various parameter counts, compared to the alternatives listed above. Full results with standard deviations are listed in Table 2, along with the number of Multiply-Accumulate (MAC) operations each network uses.

Our results show that block cheapening is more effective than simple downscaling schemes (reducing width/depth); not only are the reduced models (WRN-40-1, WRN-16-2, WRN-16-1) inflexible in parameter range, they perform significantly worse than our models at the parameter budgets they can map to. We also show that BlockSwap finds more accurate networks than the other top-down approaches ($\ell$1-pruning and SNIP) across the full parameter spectrum. Note that as $\ell$1-pruning and SNIP introduce unstructured sparsity, the parameter counts provided are the number of non-zero parameters (the weight tensors remain the same size as the original teacher).

The mixed block architectures that BlockSwap generates are more accurate at these reduced budgets than all of the single blocktype alternatives we considered. While the performance of DARTS is very similar to BlockSwap, it is worth noting that the DARTS architecture search required 24 GPU hours. By comparison, a BlockSwap search for an 800K parameter network took less than 5 minutes using a single Titan X Pascal GPU.

Given the strength of random baselines in architecture search settings (Liu et al., 2019a; Yu et al., 2020; Li & Talwalkar, 2019), we also compare BlockSwap against randomly generated mixed-blocktype configurations in Appendix C. BlockSwap consistently outperforms these, demonstrating that our Fisher potential metric is effective at selecting potent block structures.

Table 2: CIFAR-10 top-1 test error for student nets, with parameter count (in thousands, as P.(K)) and total MAC operations (in millions, as Ops(M)). D-W specifies the number of layers and the width multiplier of the student. We compare BlockSwap to reductions that rely on reducing depth (D) and width (W), single-blocktype networks (MBConv6, DARTS, CondenseNet), and pruning via SNIP (Lee et al., 2019). Comparisons to random configurations and $\ell 1$-pruning are given in Appendix C. We do not report Ops for SNIP since this is dependent on the choice of sparse representation format. BlockSwap is able to choose the networks with the lowest mean error for all parameter budgets considered.

| D-W | Method | P. (K) | Ops (M) | Err. $\mu \pm \sigma$ | D-W | Method | P. (K) | Ops (M) | Err. $\mu \pm \sigma$ |
|---|---|---|---|---|---|---|---|---|---|
| 40-2 | Teacher | 2243.5 | 328.3 | $3.96 \pm 0.09$ | 40-2 | SNIP | 404.2 | - | $4.67 \pm 0.11$ |
| 16-2 | D-Scaled | 691.7 | 101.4 | $4.78 \pm 0.13$ | 40-2 | SNIP | 289.2 | - | $5.04 \pm 0.06$ |
| 40-1 | W-Scaled | 563.9 | 83.6 | $4.57 \pm 0.07$ | 40-2 | SNIP | 217.0 | - | $5.53 \pm 0.08$ |
| 16-1 | D-W-Scaled | 175.1 | 26.8 | $7.32 \pm 0.02$ | 40-2 | SNIP | 162.2 | - | $6.00 \pm 0.16$ |
| 40-2 | MBConv6 | 1500.9 | 231.7 | $5.32 \pm 0.04$ | 40-2 | BlockSwap | 811.4 | 132.5 | $3.79 \pm 0.01$ |
| 40-2 | DARTS | 321.5 | 52.8 | $4.49 \pm 0.03$ | 40-2 | BlockSwap | 556.0 | 89.5 | $4.17 \pm 0.22$ |
| 86- | CondenseNet | 520.2 | 65.2 | $4.95 \pm 0.05$ | 40-2 | BlockSwap | 404.2 | 92.8 | $4.21 \pm 0.13$ |
| | | | | | 40-2 | BlockSwap | 289.2 | 65.9 | $4.45 \pm 0.18$ |
| 40-2 | SNIP | 811.4 | - | $4.13 \pm 0.13$ | 40-2 | BlockSwap | 217.0 | 38.8 | $4.68 \pm 0.37$ |
| 40-2 | SNIP | 556.0 | - | $4.32 \pm 0.10$ | 40-2 | BlockSwap | 162.2 | 33.9 | $5.17 \pm 0.00$ |

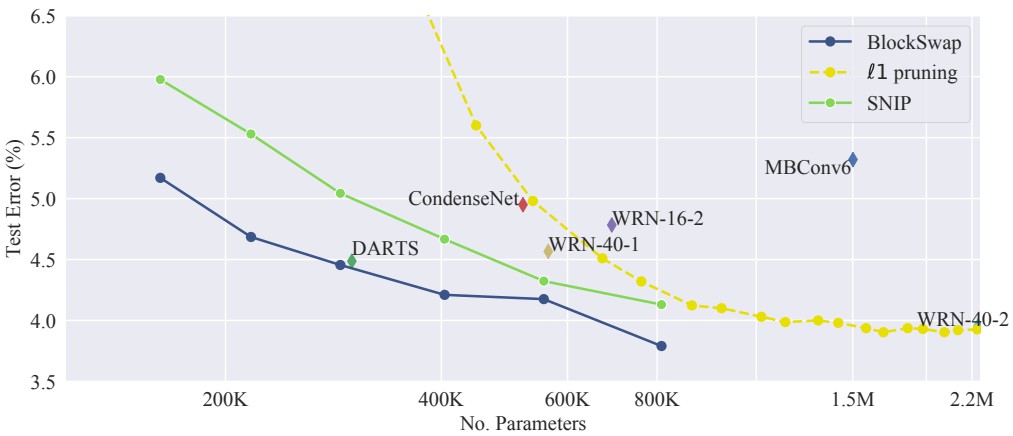

Figure 3: CIFAR-10 top-1 test error of students versus parameters. BlockSwap models (blue) give lower error for each parameter budget when compared to depth/width reduced or pruned models. They also outperform single blocktype networks (MBConv, DARTS, CondenseNet-86). Note that all networks have been trained using attention transfer, or have been *born-again* (Furlanello et al., 2018) in the case of CondenseNet.

**Implementation Details:** Networks are trained for 200 epochs using SGD with momentum 0.9. The initial learning rate of 0.1 is cosine annealed (Loshchilov & Hutter, 2017) to zero across the training run. Minibatches of size 128 are used with standard crop + flip data augmentation and Cutout (DeVries & Taylor, 2017). The weight decay factor is set to 0.0005. For attention transfer $\beta$ is set to 1000.

## 6 IMAGENET CLASSIFICATION

Here, we demonstrate that students chosen by BlockSwap succeed on the more challenging ImageNet dataset (Russakovsky et al., 2015). We use a pretrained ResNet-34 (16 blocks, 21.8M parameters) as a teacher, and compare students at two parameter budgets (3M and 8M). We train a BlockSwap student

at each of these budgets and compare their validation errors to those of a reduced depth/width student (ResNet18 and ResNet-18-0.5—a ResNet-18 where the channel width in the last 3 sections has been halved) and a single-blocktype student (ResNet-34 with G(4) and G(N) blocks). The student networks found by BlockSwap for these two budgets are illustrated in Appendix D. Top-1 and top-5 validation errors are presented in Table 3. At both budgets, BlockSwap chooses networks that outperform its comparators. At 8M parameters it even *surpasses the teacher* by quite a margin. Specifically, it beats the teacher by 0.49% in top-1 error and 0.82% in top-5 error despite using almost $3\times$ fewer parameters.

**Implementation Details:** Networks are trained with a cross-entropy loss for 100 epochs using SGD with momentum 0.9. The initial learning rate of 0.1 is reduced by $10\times$ every 30 epochs. Minibatches of size 256—split across 4 GPUs—are used with standard crop + flip augmentation. The weight decay factor is set to 0.0001. For attention transfer $\beta$ is set to 750 using the output of each of the four sections of network.

Table 3: Top-1 and Top-5 classification errors (%) on the validation set of ImageNet for students trained with attention transfer from a ResNet-34. We can see that for a similar number of parameters, the student found from BlockSwap outperforms its counterparts, and in one instance, the teacher.

| Model | Params | MACs | Top-1 err | Top-5 err |
|---|---|---|---|---|
| ResNet-34 **Teacher** | 21.8M | 3.669G | 26.73 | 8.57 |
| ResNet-18 | 11.7M | 1.818G | 29.18 | 10.05 |
| ResNet-34-G(4) | 8.1M | 1.395G | 26.58 | 8.43 |
| BlockSwap | 8.1M | 1.242G | **26.24** | **7.75** |
| ResNet-18-0.5 | 3.2M | 909M | 37.20 | 15.02 |
| ResNet-34-G(N) | 3.1M | 559M | 30.16 | 10.66 |
| BlockSwap | 3.1M | 812M | **29.57** | **10.20** |

## 7 COCO DETECTION

Thus far, we have used BlockSwap for image classification problems. Here we observe whether it extends to object detection on the COCO dataset (Lin et al., 2014)—specifically, training on 2017 train, and evaluating on 2017 val. We consider a Mask R-CNN (He et al., 2017) with a ResNet-34 backbone, and apply BlockSwap using COCO images to obtain a mixed-blocktype backbone with 3M parameters. We compare this to a single-blocktype ResNet-34-G(N) backbone which uses the same number of parameters. To avoid conflation with ImageNet, we train everything **from scratch**. The results can be found in Table 4. We can see that the BlockSwap backbone again outperforms its single-blocktype counterpart.

**Implementation Details:** Networks are trained using the default Mask R-CNN settings in Torchvision. We use a batch-size of 16 split across 8 GPUs. All models are trained from scratch, and we forgo distillation due to memory constraints.

Table 4: Average Precisions (%) for COCO-2017 val detection for Mask R-CNNs using a ResNet-34-G(N) and BlockSwap backbone (each using 3M parameters).

| Backbone | $AP$ | $AP_{50}$ | $AP_{75}$ | $AP_S$ | $AP_M$ | $AP_L$ |
|---|---|---|---|---|---|---|
| ResNet34-G(N) | 22.3 | 38.9 | 23.0 | 12.9 | 22.9 | 30.7 |
| BlockSwap | **23.4** | **40.0** | **24.7** | **13.6** | **24.2** | **31.1** |

## 8 CONCLUSION

We have developed BlockSwap: a fast, simple method for reducing large neural networks to flexible parameter targets based on block substitution. We have verified that these reduced networks make

for excellent students, and have performed a comprehensive ablation study. Future work could use BlockSwap to choose networks based on inference time, or energy cost instead of parameter count.

ACKNOWLEDGMENTS

This work was supported in part by the EPSRC Centre for Doctoral Training in Pervasive Parallelism and a Huawei DDMPLab Innovation Research Grant, as well as funding from the European Union's Horizon 2020 research and innovation programme under grant agreement No.732204 (Bonseyes). This work is supported by the Swiss State Secretariat for Education, Research and Innovation (SERI) under contract number 16.0159. The opinions expressed and arguments employed herein do not necessarily reflect the official views of these funding bodies. The authors are grateful to David Sterratt for his LaTeX prowess, and to the BayesWatch team and anonymous reviewers for their helpful comments.

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

# A    BLOCK REPLACEMENTS

Table 5: Block substitutions used in this paper: Conv refers to a $k \times k$ convolution. GConv is a grouped $k \times k$ convolution and Conv$1 \times 1$ is a pointwise convolution. We assume that the input to each block has $N$ channels and that channel size doesn't change unless written explicitly as $(x \rightarrow y)$. Where applicable, $g$ is the number of groups and $b$ is the bottleneck contraction. The convolutional and batch-norm (BN) costs are given, although the latter is significantly smaller. BN+ReLU precedes each convolution for WideResNets and follows each convolution for standard ResNets.

| Block | $S$ | $G(g)$ | $B(b)$ | $BG(b, g)$ |
|---|---|---|---|---|
| Structure | Conv | GConv (g) | Conv$1 \times 1(N \rightarrow \frac{N}{b})$ | Conv$1 \times 1(N \rightarrow \frac{N}{b})$ |
|  | Conv | Conv$1 \times 1$ | Conv | GConv(g) |
|  |  | GConv (g) | Conv$1 \times 1(\frac{N}{b} \rightarrow N)$ | Conv$1 \times 1(\frac{N}{b} \rightarrow N)$ |
|  |  | Conv$1 \times 1$ |  |  |
| Conv Params | $2N^2k^2$ | $2N^2(\frac{k^2}{g} + 1)$ | $N^2(\frac{k^2}{b^2} + \frac{2}{b})$ | $N^2(\frac{k^2}{gb^2} + \frac{2}{b})$ |
| BN Params | $4N$ | $8N$ | $N(2 + \frac{4}{b})$ | $N(2 + \frac{4}{b})$ |

# B    UNDERSTANDING BLOCK PLACEMENT

To understand what makes for "good" block placement, we begin with the more tractable question: if we were only to cheapen one block, which one should we choose? In Figure 4 we show what happens when we substitute a single standard block with G(4) for each of the possible blocks for our WideResNet. We plot the final test error after training to convergence, as well as the total parameter count after the substitution (the higher the bar, the higher the error, therefore the more sensitive the block).

Notice that in Figure 4(a), at substituting the start and end of each *stage* appears to have a very detrimental effect on accuracy, whereas mid-stage blocks can be substituted without hurting performance. For example, block 12 has few parameters, as shown in Figure 4(b), but substituting it for a cheaper alternative results in a drastic drop in accuracy. This implies that it is safest to substitute blocks in the middle of a stage but that blocks at the start and end of each stage should have their representational capacity preserved as much as possible.

We used this insight to analyse the block configurations of "good" and "bad" amongst the random architectures at each parameter budget. With several blocktypes, however, we need a method to rank the capacity of blocktypes; we make the simplifying assumption that training the blocktype on its own is a good indicator of this (though this is not always true, since blocks give varying performance with different input sizes). An example of a good and bad network judged this way can be seen in Figure 5; while they do roughly follow our observations, there are clearly more complex interactions

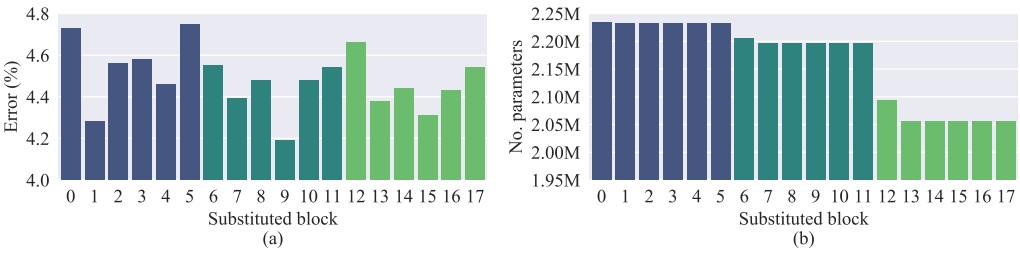

Figure 4: Each bar shows a WideResNet-40-2 with one block substituted for G(4), where (a) shows final CIFAR-10 error and (b) shows resulting number of parameters, with blocks coloured by network stage. This shows us the sensitivity of each block; for example, block 12 has relatively few parameters, but substituting it for a cheaper block is very detrimental to final error.

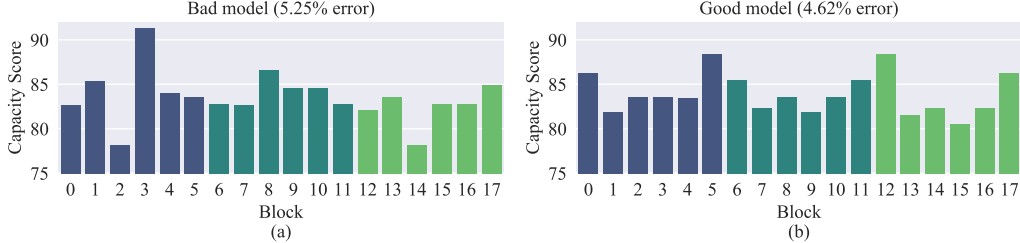

Figure 5: An example of (a) a bad architecture and (b) a good architecture for a given parameter budget (200K), with blocks coloured by network stage. Capacity score is given by the accuracy that the selected blocktype achieves after being trained on its own. We can see that the good model (b) follows the pattern observed in Figure 4, while the bad model (a) does not.

at play (perhaps unsurprising given the random nature of their construction). We found that overall, the models in our ablation set adhere to the pattern described above (for each stage: high capacity blocks at the start and end, increasing capacity throughout), with some variance.

## C    COMPARING TO RANDOM CONFIGURATIONS AND $\ell$1-PRUNING

Table 6: Comparing BlockSwap to randomly configured mixed-blocktype networks and $\ell$1-pruned versions of the original teacher on the CIFAR-10 dataset. Total MAC operations are not reported for $\ell$1-pruning because calculating this is highly dependent on choice of sparse tensor representation format.

| D-W | Method | Parameters (K) | MAC Ops (M) | Error ($\mu \pm \sigma$) |
|------|-------------|----------------|-------------|----------------------|
| 40-2 | BlockSwap | 811.4 | 132.5 | $3.79 \pm 0.01$ |
| 40-2 | BlockSwap | 556.0 | 89.5 | $4.17 \pm 0.22$ |
| 40-2 | BlockSwap | 404.2 | 92.8 | $4.21 \pm 0.13$ |
| 40-2 | BlockSwap | 289.2 | 65.9 | $4.45 \pm 0.18$ |
| 40-2 | BlockSwap | 217.0 | 38.8 | $4.68 \pm 0.37$ |
| 40-2 | BlockSwap | 162.2 | 33.9 | $5.17 \pm 0.00$ |
| 40-2 | Random | 795.7 | 81.6 | $4.26 \pm 0.04$ |
| 40-2 | Random | 551.1 | 76.7 | $4.54 \pm 0.18$ |
| 40-2 | Random | 397.4 | 59.5 | $4.91 \pm 0.12$ |
| 40-2 | Random | 285.1 | 56.7 | $4.80 \pm 0.12$ |
| 40-2 | Random | 217.6 | 46.9 | $5.13 \pm 0.24$ |
| 40-2 | Random | 168.3 | 33.6 | $5.75 \pm 0.09$ |
| 40-2 | $\ell$1-pruning | 894.7 | - | $4.12 \pm 0.03$ |
| 40-2 | $\ell$1-pruning | 760.5 | - | $4.32 \pm 0.06$ |
| 40-2 | $\ell$1-pruning | 671.1 | - | $4.51 \pm 0.13$ |
| 40-2 | $\ell$1-pruning | 536.8 | - | $4.98 \pm 0.21$ |
| 40-2 | $\ell$1-pruning | 447.4 | - | $5.60 \pm 0.23$ |
| 40-2 | $\ell$1-pruning | 313.2 | - | $7.70 \pm 0.18$ |
| 40-2 | $\ell$1-pruning | 223.7 | - | $11.93 \pm 0.77$ |

# D    BLOCKSWAP NETWORKS FOR IMAGENET

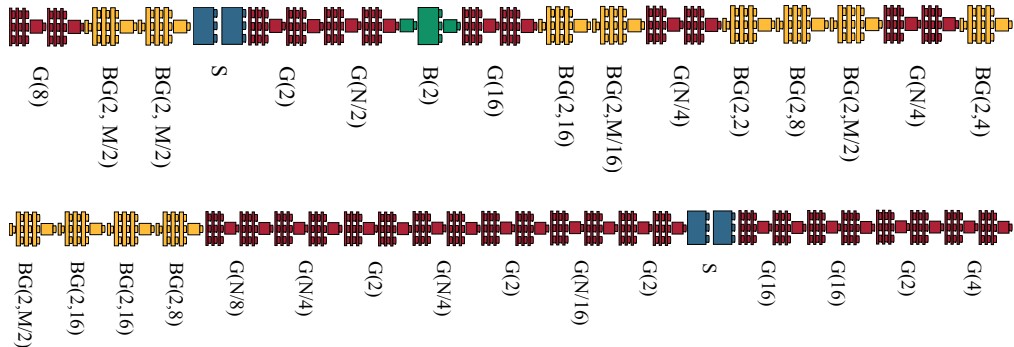

Figure 6: ResNet-34 block substitutions chosen by BlockSwap at 3M (top) and 8M (bottom) parameters on ImageNet.

