# OpenReview forum: "BlockSwap: Fisher-guided Block Substitution for Network Compression on a Budget"
_ICLR.cc/2020/Conference — Accept (Poster)_

### Official Review · AnonReviewer3 · 2019-10-17
**Official Blind Review #3**

**Rating:** 6

**Review:**

This paper introduces an approach to compressing a deep network so as to satisfy a given budget. In contrast to methods that replace all convolutional blocks with the same cheaper alternative, the authors argue that mixing different types of such cheap alternatives should be effective. To achieve this in a fast manner, they propose to randomly sample architectures with mixed blocktypes that satisfy the given budget and rank these architectures using Fisher information.

Originality:
- The approach is simple, but seems effective. Although the different components that make up the final method are known, they are put together so as to address a different task, in a way that I find interesting.

Methodology:
- From the paper, it is not entirely clear how sampling under budget constraints is achieved. I imagine that one could do it in a naive way, by choosing the first block uniformly randomly, and then removing the candidates that do not fit the budget for the subsequent blocks. This, however, would seem to give a different importance to the early blocks than to the later ones, since the former would essentially not have any budget constraints. I would appreciate it is the authors could explain their strategy in more detail.
- The assumption that the number of blocks in the student is the same as in the teacher seems constraining. A better architecture might be obtained by having fewer, wider blocks, or more, thinner blocks. Do the authors see a potential way to address this limitation?
- In principle, it seems that the proposed Fisher Information-based strategy could be used for general NAS, not just for compact architectures. Have the authors investigated this direction?

Related work:
- It seems to me that the literature review on compression/pruning is a bit shallow. I acknowledge, however, that most works do not tackle the scenario where a budget is given. However, Chen et al., "Constraint-aware Deep Neural Network Compression", ECCV 2018, do, and it would be interesting to discuss and provide comparisons with this work.

Experiments:
- I appreciate the ablation study, which answered several of my questions.
- In Table 1, it seems counterintuitive that the (negative) correlation becomes smaller as the number of mini-batches increases. Do the authors have an explanation for this?
- In Section 5, are the baseline compact networks all trained using the same Attention Transfer algorithm as for the BlockSwap ones?
- In Table 2, the budget values (P. (K)) seem fairly arbitrary? How were they obtained? They seem to match those of SNIP. Is this because the authors ran SNIP, and then set their budget accordingly?
- Below Fig. 3, the authors mention that they are generous with sparsity-based methods because they count the number of non-zero parameters. Note that several structured-sparsity compression methods have been proposed (Alvarez & Salzmann, NIPS 2016, 2017; Wen et al., NIPS 2016), which, by contrast with regular sparsity ones, would cancel out entire filters.

Summary:
Overall, I like the simple idea proposed in this paper. I would however appreciate it if the authors could clarify the way they sample the architectures and address my questions about their experiments.

**Experience Assessment:**

I have published one or two papers in this area.

**Review Assessment: Checking Correctness Of Derivations And Theory:**

I carefully checked the derivations and theory.

**Review Assessment: Checking Correctness Of Experiments:**

I carefully checked the experiments.

**Review Assessment: Thoroughness In Paper Reading:**

I read the paper thoroughly.

---

> ### Author Response · Authors · 2019-11-07
> **Response to Reviewer #3**
>
> We would first like to thank the reviewer for their detailed comments and analysis of our work. We are glad they liked the idea, and we are particularly happy that they appreciated the simplicity of the method, as we believe that complicated methods can present a major obstacle to deployment.
>
> We answer the queries below:
>
> - The sampling under budget constraints is done using rejection sampling: we generate architecture proposals at random and only save those that satisfy our parameter budget. Since we are only generating proposals there is no need to create or instantiate any networks, we are able to infer the parameter budget directly from the configuration, making this a very cheap operation. We will make this clear in the paper. This could of course be improved in future, perhaps using a constraint solver to generate possible networks, or learning a distribution over block types.
>
> - The suggestion that student networks be allowed to vary in number of blocks is a very interesting one, and not one that we had considered. We worry that comparing Fisher potential across different depth networks may raise complications. However, one means to achieve this might be to introduce “Identity” as a block choice, which would allow for depth shrinking/expansion.
>
> - Regarding the use of Fisher information for general NAS: this also is an interesting suggestion, and something we plan to investigate in future work.
>
> - Thank you for pointing out the related work by Chen et al. Their statement of budget constraints as a problem of Bayesian optimisation is very interesting. We will add this to our related work and provide some discussion.
>
> - On the reduced correlation of Fisher potential with final performance after training with several minibatches of data: At the moment we can only speculate on this. At initialisation, with randomly sampled weights, the activations may be more conditionally independent than later in training, when the weights are being learnt. The use of the Fisher potential implicitly assumes conditional independence as a measure of the information a block has about the label.
>
> - We can also confirm that each baseline network is trained with the same Attention Transfer mechanism as the BlockSwap ones, or with a “born again” strategy [1] in the case of CondenseNet. We will amend the figure caption to make this clear.
>
> - The budgets we tested with were chosen to be log-linearly spread based on the parameter count of the original model (WRN-40-2). We then also chose SNIP budgets to match.
>
> - Regarding sparsity and non-zero parameter counts; we were specifically comparing with unstructured sparsity approaches, due to its relationship with SNIP. We will make this more explicit in the paper.
>
> We hope that this answers all of the questions the reviewer raised and again thank them for such constructive feedback.
>
> [1] Furlanello et al. Born Again Neural Networks, ICML 2018

---

### Official Review · AnonReviewer1 · 2019-10-23
**Official Blind Review #1**

**Rating:** 3

**Review:**

1) in the introduction, it's not clear what attention transfer means
  2) why did they specifically choose these four blocks?
  3) (3.1) substitute blocks $S$: ``each using $N$ lots of $N \times k \times k$ filters" should it be ``$N$ number of $k \times k$ filters"?
  4) (3.2) ``consider a choice of layers $i = 1, 2, \dots, N_L$" what is $L$ and why does the number of layers depend on $L$?
  5)  (3.2) Why is that specific $f$ chosen?
  6)  (3.3) They could elaborate more on why they chose Fisher potential instead of other metrics for architecture selection. Currently, they only provided the intuition of the metric. Since the paper suggests the Fisher potential is a crucial part of their method, they could provide more theoretical justification about this choice.
  7) (4.0) How did they decide on the hyperparameters?
  8) In the introduction, they suggest that the major novelty of their method ``assigns non-uniform importance to each block by cheapening them to different extents", but in their method they only randomly assembled the mixed-blocktype models.
  9) Section 4.1 suggests good mixed blocktype yields low error but they didn't address how good mixed blocktype can be found

**Experience Assessment:**

I have published one or two papers in this area.

**Review Assessment: Checking Correctness Of Derivations And Theory:**

I assessed the sensibility of the derivations and theory.

**Review Assessment: Checking Correctness Of Experiments:**

I assessed the sensibility of the experiments.

**Review Assessment: Thoroughness In Paper Reading:**

I read the paper thoroughly.

---

> ### Author Response · Authors · 2019-11-07
> **Response to Reviewer #1**
>
> We would like to thank the reviewer for their questions, and we are very happy to answer them. We hope this reiterates the important point of this paper: it shows how good mixed blocktypes can be found and extensively demonstrates their superior performance.
>
> 1. Attention transfer is the term for the widely-used distillation technique we employ. We cite the original paper [1] in the introduction on the first use of the term so hopefully readers who are unfamiliar with the method can follow up on it. We provide a formal and informal definition in Section 3.2 for completeness. We will explicitly add that attention transfer is a distillation method in the introduction.
>
> 2. The blocks chosen are a representative subset of those that have previously been demonstrated to provide substantial compressive advantage. The Standard (S) block is used in most residual networks and the remaining blocks are cheapened versions of S. The Bottleneck (B) block is very popular in compact network designs. Grouped convolutions (as seen in G) are prevalent in many mobile-sized networks (e.g. MobileNet, CondenseNet), and the BG block is simply a combination of B and G. These blocks are not highly-engineered, so we can demonstrate that the success of our method is due to their combination, rather than their raw representational capacity. The fact we can make gains from using this small selection of simple blocks is itself enlightening.
>
> 3. Thank you for checking our numbers regarding the convolutions in the S block.  In this instance, we are correct in the paper: if the number of input channels to a convolution is $N$, and the number of output channels is also $N$, then each filter of kernel size $k\times k$ has depth N. There are N of these $N\times k\times k$ filters, each one producing one of the N output channels.
>
> 4. $N_L$ can be read as the number of layers in the network. L is not itself an object, just a notation. We will change the notation and writing here to make things very clear to prevent others from being distracted by this.
>
> 5. We mention in Section 3.2 that the attention transfer mapping f() is taken directly from [1]. It is critical that we use the common default attention transfer mapping so any demonstrated benefit is not arguably down to some selection of f(), but due to the BlockSwap procedure.
>
> 6. Fisher potential is theoretically motivated: the Fisher potential is the trace of the Fisher Information matrix of the activations. In each block the Fisher potential is (locally) the total information the block contains about the class label (under a simplifying conditional independence assumption). During training it is this information about the class that drives learning, and the initial learning steps are key. Hence higher information values tend to result in higher efficiency block utilisation. It is interesting to note that at initialisation, with randomly sampled weights, the activations may (speculatively) be more conditionally independent than later in training, and this may explain the benefit of doing this computation on the first minibatch. In future we will explore this further. In addition to the theoretical motivation, we empirically show Fisher potential to be better than common, less well-founded alternatives in Table 1.
>
> 7. We chose our training hyperparameters to closely mirror those detailed in the original WideResNet paper [2] for CIFAR to ensure the empirical results focus on the BlockSwap procedure. All the design hyperparameters for BlockSwap (e.g. number of samples taken) are justified in Section 4.
>
> 8. The major novelty of this paper is a very efficient method that determines which of a large number of mixed blocktype networks---in which the blocks considered are cheaper versions of the original S block---is a powerful network. It is more important to cheapen the structure in some blocks of the network than others. This relative importance is found via search, rejecting structures with the wrong balance using the Fisher potential.
>
> 9. In fact, the primary contribution of this paper is precisely that it introduces an algorithm that finds a good mixed blocktype. This is done through search across randomly generated blocks that satisfy a budget for a mixed block network that optimizes a very cheap Fisher potential calculation. Because the Fisher potential is cheap to calculate (it does not involve learning) and correlates very well with learnt network performance, a large number of random networks can be tested, and hence we can output very high performing networks. We provide substantial empirical evidence that our method for finding such networks outperforms randomly assembling them. We also provide analyses of exactly what makes a good architecture in Appendix B.
>
> [1] Zagoruyko and Komodakis, Paying More Attention to Attention: Improving the Performance of Convolutional Neural networks via Attention Transfer. ICLR, 2017
>
> [2] Zagoruyko and Komodakis, Wide residual networks, BMVC 2016

---

### Official Review · AnonReviewer2 · 2019-10-25
**Official Blind Review #2**

**Rating:** 6

**Review:**

This paper studies a fast algorithm for choosing networks with interleaved block types – BlockSwap. It passes a single minibatch of training data through randomly initialized networks and gauging their Fisher potential.
The teacher-student network is used here to learn compressed network on a budget. They conduct various experiments, including cifar-10, imagenet, and coco-detection. All the experiments show the advance of proposed model, which is quite remarkable.
Overall, this paper is well organized, and very well written. The insightful experiments thoroughly discuss and compare the proposed method.


**Experience Assessment:**

I have read many papers in this area.

**Review Assessment: Checking Correctness Of Derivations And Theory:**

I assessed the sensibility of the derivations and theory.

**Review Assessment: Checking Correctness Of Experiments:**

I assessed the sensibility of the experiments.

**Review Assessment: Thoroughness In Paper Reading:**

I made a quick assessment of this paper.

---

> ### Author Response · Authors · 2019-11-07
> **Response to Reviewer #2**
>
> We would like to thank the reviewer for their positive review. We are delighted that they found the results remarkable, and the experiments insightful and thorough.

---

### Decision · Program_Chairs · 2019-12-19

**Decision:**

Accept (Poster)

**Comment:**

Two reviewers recommend acceptance. One reviewer is negative, however, does not provide reasons for rejection. The AC read the paper and agrees with the positive reviewers. in that the paper provides value for the community on an important topic of network compression.